# Vascular Alterations in Uterine and Ovarian Hemodynamics and Hormonal Analysis throughout Pregnancy Loss in Cows under Heat Stress

**DOI:** 10.3390/vetsci11100479

**Published:** 2024-10-05

**Authors:** Elshymaa A. Abdelnaby, Abdulrhman K. Alhaider, Ibrahim M. Ghoneim, Ibrahim A. Emam

**Affiliations:** 1Department of Clinical Sciences, College of Veterinary Medicine, King Faisal University, P.O. Box 400, Al-Ahsa 31982, Saudi Arabia; 2Theriogenology Department, Faculty of Veterinary Medicine, Cairo University, Giza 12211, Egypt; 3Department of Surgery, and Radiology, Faculty of Veterinary Medicine, Cairo University, Giza 12211, Egypt

**Keywords:** cows, Doppler, nitric oxide, progesterone, RI

## Abstract

**Simple Summary:**

The ovarian and middle uterine artery blood flows were examined in cows under heat stress conditions regarding hormonal profile. Luteal vascularity was declined in cows with embryonic death. Progesterone levels elevated in cows with embryonic death, then declined. In addition, both Doppler indices were elevated in cows that suffered from embryonic death. This study provided facts about the relations among the luteal diameter and luteal hemodynamics that predict the amount of blood supply, which act as a sensitive parameter to detect the alterations in luteal function during the first 50 days after mating.

**Abstract:**

This current study examined the ovarian (OA) and middle uterine arteries (MUA) blood flow under heat stress conditions regarding hormonal status (progesterone (P4), estradiol (E2), and nitric oxide metabolites (NOMs) assays). Eighteen pluriparous cows were examined, with twelve only being subjected to the natural mating as the other six animals were not bred. Pregnancy diagnosis was confirmed at day 30 by embryonic heartbeat and CL graviditatis (*n* = 6; pregnant), but some animals (*n* = 6) showed early embryonic death (EED), with the presence of control cows (*n* = 6). In the pregnant group, luteal diameter (cm) increased after mating, while in the pregnancy loss group it increased (*p* < 0.05) until reaching day 38 (1.41 ± 0.03), then decreased again. Luteal vascularity was declined in cows with EED after day 36 (*p* < 0.05) and reached its lowest level at day 50. P4 levels elevated in cows with EED until day 36 (13.64 ± 0.11) then declined. Both ipsilateral OA and MUA Doppler indices were declined in both groups except in cows who suffered from EED; both were elevated from day 38 until day 50 after mating. Ipsilateral peak systolic point (PSV cm/sec) of OA and MUA was elevated in both groups (*p* < 0.05), but in cows with EED this parameter was declined. E2 and NOMs levels were declined in cows with EED from day 40 and day 38 after mating (*p* < 0.05), respectively. This study provided novel data on the relations among the luteal diameter, E2, P4, and NOM levels, and luteal hemodynamics that predicts the amount of blood supply, which acts as a sensitive parameter to detect the alterations in luteal function during the first 50 days after mating.

## 1. Introduction

Most producers and farmers of bovine species face problems that influence financial returns, one of which is reproductive bovine efficiency [1]. Reproductive efficiency is the main component of economic achievement in dairy herds [2]. Methods for increasing the number of calves and heifers produced have been developed [3], but fetal losses and the premature death of developing embryos provide an overwhelming obstacle. Cows at risk of miscarriage can be identified using a variety of techniques, which may present opportunities to develop intervention programs that will benefit those harmed [4]. The constant exposure to high humidity and warmth in the environment had an impact on steroidogenesis as well as oocyte viability [5], in addition to reducing the oocyte quality and fertilization rate [6,7,8]. A decline by 33% in oocyte competence was observed in cows exposed to heat stress (HS) [9]. HS also prevents oocyte growth [10] and impairs oocyte maturation and embryonic development [11,12,13]. Estradiol levels were declined in the summer season, which was affected negatively by the estrous sign manifestations [14], while the mean prolactin levels were elevated, which adversely affected the buffalo infertility due to acyclicity [13]. By affecting the quality of the oocyte, the activity of the follicles, and the levels of blood plasma progesterone (P4), summer heat stress reduces the fertility in cattle in hot locations [13]. However, it is still unknown how increased temperatures affect how well the corpus luteum (CL) functions. As low as 25–27 °C is the upper critical temperature—the ambient temperature below which nursing dairy cows cannot maintain a steady body temperature [11]. As a result, heat stress affects animals in addition to its significant impact on the price of milk and beef [10].

In dairy cattle, abnormalities of the embryo and fetus found by ultrasound are accurate indicators of either pregnancy failure or the outcome of the pregnancy with a normal birth [15].Hemodynamic function may also be a risk factor for predicting the preservation or loss of a cow pregnancy, in addition to the corpus luteum (CL); however, the CL diameter alone cannot predict the maintenance or loss of the embryo during pregnancy [16,17]. Progesterone (P4) levels are also a reliable indicator of pregnancy loss [17,18]. In cyclic and pregnant cattle, a positive association between P4 levels and CL blood flow was investigated using Doppler devices [19,20]. This Doppler method was first applied to Holstein Friesian cows’ induced [21] and spontaneous [22] luteolysis in order to measure the real-time fluctuations in luteal blood flow status of coloring. Additionally, during the estrous cycle of mares [23] and Holstein cows [19], Doppler was utilized to evaluate the luteal flow with luteal size (LS) and concentrations of P4.

Alterations in the vascular resistance (RI) in the uterine artery ipsilateral to the embryo compared with the contralateral uterine artery were distinguished in pregnant cattle at 60 d of pregnancy [24] and as early as d 16 of pregnancy in heifers [25]. There are few women reports about changes in uterine and ovarian flow in pregnancy loss [26]. There is little information related to luteal dynamics vascularization linked to a reduced blood flow rate based on luteal blood supply in relation to hemodynamic changes in the ovarian and uterine arteries between pregnancy and pregnancy loss in dairy Holstein cows. Therefore, this study estimated whether ipsilateral ovarian or uterine artery vascular dynamics would recognize cows at risk of pregnancy loss due to the early death of the newly developing embryo. The hypothesis was that cows that abort after d 38 would have decreased CL perfusion detected by the increase in Doppler indices (suggestive of decreased uterine blood flow) and the decline in peak systolic velocity of both ovarian and uterine arteries.

## 2. Materials and Methods

### 2.1. Cow Housing

Eighteen lactating pluriparous Holstein adult cows (*n* = 18) of 7–9 years old with a body condition score (BCS) of 4 ± 0.5 and weighing 600 ± 50 kg were kept under maintenance conditions of feeding and management in the faculty research farm for large animals. Cows’ maintenance requirements for nutrition consist of concentrated rations, green fodder (40 kg), and hay with free access to water. The reproductive assessment of cows was performed using an ultrasound in addition to signs after mating. This study was accepted by the Faculty of Veterinary Medicine at King Faisal University, which approved experimental protocols with an approval number of KFU-2024-ETHICS 2625. The cows’ health is preserved by clean housing, clean water, a balanced diet, and taking the appropriate precautions against common diseases.

### 2.2. Study Design

The assessment of animals was in the hottest months from July to September 2022 with a temperature humidity index (THI) greater than 72 (Table 1), as THI was calculated by (1.80 × T + 32) − (0.55 − 0.0055 × RH) × (1.80 × T − 26) as T is temperature and RH is relative humidity. Animals were examined twice daily to regulate the onset of estrus (day 0). B-mode gray normal ultrasound was achieved to check the existence of a preovulatory follicle. Cows were examined (*n* = 18) by ultrasound, and then the experimental design was as follows: cows served as a relevant control in this study that was open (not bred; *n* = 6) cows to assess the timing of luteal regression and vascular conditions in this group relative to the other groups, and then the other cows (*n* = 12) were mated naturally using the adult mature bull with proven fertility 10–12 hrs after the onset of standing estrous. B-mode ultrasonography was accomplished the following day after mating to confirm the absence of POF (ovulation). From all mated cows, only six became pregnant by evaluating the CL of pregnancy from day 10 to day 22 (*n* = 6) with heartbeat at day 30 [23], but some animals showed pregnancy loss and suffered from early embryonic death after day 38 (EED, *n* = 6) that confirmed the decline in the plasma levels of pregnancy-specific protein B (PSPB) that confirms the early death of the conceptus. To compare the luteal diameter and luteal blood flow indicated by luteal colored area, which related to the amount of vascular perfusion after mating, B- and color modes of the ovaries were performed from day 10 to day 50 in control (*n* = 6), pregnant (*n* = 6), and cows with EED (*n* = 6). During this period, Doppler ultrasonographic examinations were carried out day after day using color and spectral waves. There were no comparisons made between cows in heat stress and thermoneutral conditions, but the THI was taken as a factor only.

### 2.3. B- and Color Modes for Luteal Changes and Uteroovarian Vascular Perfusion

Both ovaries were imaged in B-mode utilizing a 6 MHz linear array transrectal transducer. The same operator performed all of the ultrasound examinations, and it took roughly ten minutes for each animal. Three cross-sectional pictures were used to measure the greatest area of the corpus luteum (CL). To validate the quantity of colored area/pixels, the Color Doppler mode was employed. A trans-rectal transducer (EXAGO, Meyreuil, France) with the following specifications was used: Power: Max. (100%); PRF: 4 kHz; Doppler angle: 45°; and Brightness: 78% (Figure 1).

### 2.4. Data and Image Analysis

The diameter of each corpus luteum (CL) or follicle within each ovary, ovarian, and uterine arteries was measured using computerized ultrasonography calipers [23]. Peak systolic velocity (PSV), resistance index (RI), and pulsatility index (PI) were all assessed using the spectral Doppler [27]. The luteal color flows were calculated per pixel using Adobe Photoshop [28].

### 2.5. Sampling and Hormonal Assaying

Blood was collected from the jugular vein from day 10 to day 50 in the early morning by an 18-gauge needle. Estradiol and progesterone were analyzed using a commercially available ELISA kit (DRG, Germany; K030-H is a catalog number for estradiol, and the number for progesterone is 6107620 (Competitive EIA)) as previously measured [29]. The intra- and inter-assay coefficients of variation were (2.71 and 9.39%, 6.86 and 5.59), respectively [30]. For measuring NOMs, serum samples were mixed with Griess reagent and incubated. The sensitivity of the assay was 0.225 µmol/L nitrites in the sample [31]. Serum PSPB levels were measured by the radioimmunoassay method of double antibody [32] with catalog number CSB-E13353B. Assay sensitivity was 0.4 pg/mL for estradiol. All hormones are measured in a process of correct labeling, correct sampling, and the correct amount of sample with a perfect transport (pre-analytical phase), then all samples were measured with a correct selection of the test regarding the biosafety measures (analytical), and finally data were recorded in a correct report with an interpretation (post-analytical phase).

### 2.6. Statistical Analysis

Descriptive statistics are presented as means and standard errors of the mean (SEM). Mean values of luteal diameter, progesterone, estradiol, nitric oxide, luteal colored area, ovarian and uterine artery Doppler indices, and their peak velocity were subjected to analysis of variance (ANOVA) of repeated measurements using a general linear model to study the effect of the days on ovarian and uterine artery hemodynamics in pregnant cows and those with early embryonic death (EED) using SPSS software (2007). Data are presented in plots with error bars. Pearson correlations between the ipsilateral ovarian and middle uterine artery Doppler indices in the pregnant and pregnancy loss groups due to early embryonic death are also reported. 

## 3. Results

### 3.1. Luteal Size, Colored Area/Pixels, and Progesterone Levels

Days significantly (*p* < 0.05) affected luteal diameter (cm), colored area (pixels), and plasma levels of progesterone hormone (ng/mL). In pregnant cows, luteal diameter (cm) increased linearly until day 48 (1.91 ± 0.01) after mating, while in cows with EED, the luteal diameter (cm) increased until day 38 (1.41 ± 0.03), then decreased again after that until reaching minimum levels at day 50 (Figure 2a) compared to other groups. Luteal vascularity indicated by the amount of colored area (pixels) was declined in cows with EED after day 36 and reached its lowest level at day 50 (Figure 2b). Also, the plasma levels of progesterone were elevated in cows with EED until day 36 (13.64 ± 0.11) then declined after that compared to the levels in normal control and pregnant cows (Figure 2c). There were group, time, and group*time interaction effects (*p* < 0.05, *p*-values were equal for all the effects). The levels of pregnancy-specific protein b (PSPB) were decreased significantly in the pregnancy loss group compared to the pregnant normal one (Table 2).

### 3.2. Ipsilateral Middle Uterine (MUA) and Ovarian Arteries (OA) Doppler Indices

Both Doppler indices expressed by resistance and pulsatility indices (RI and PI) were significantly (*p* < 0.05) declined in both groups except in cows suffering from EED; both parameters of the ipsilateral side of the middle uterine artery (MUA) were elevated from day 38 until day 50 after mating compared to control and pregnant cows (Figure 3). Both Doppler indices of the ipsilateral ovarian artery (OA) were significantly (*p* < 0.05) declined in both groups except in cows suffering from EED; both indices were elevated from day 38 until day 50 after mating compared to control and pregnant cows (Figure 4). There were group, time, and group*time interaction effects (*p* < 0.05, *p*-values were equal for all the effects).

### 3.3. Doppler Peak Velocity in Both MUA and OA

Ipsilateral peak systolic velocity point (PSV cm/sec) of the OA and MUA was significantly (*p* < 0.05) elevated in both groups, but in cows with EED, this parameter was declined from day 38 until day 50 after mating (Figure 5). Ipsilateral MUA PI in the pregnant group is positively correlated with ipsilateral MUA RI, OA RI, and OA PI in the same group but negatively correlated with the parameters of the pregnancy loss group as ipsilateral OA and MUA Doppler indices. The ipsilateral OA PI in the pregnant group is positively correlated with ipsilateral RI and negatively correlated with the parameters of the pregnancy loss group as ipsilateral OA Doppler indices (Table 3). There were group, time, and group*time interaction effects (*p* < 0.05; *p*-values were equal for all the effects).

### 3.4. Estradiol (E2) and Nitric Oxide Metabolites (NOMs)

Estradiol 17ß (pg/mL) levels were significantly (*p* < 0.05) declined in cows with EED from day 40 after mating compared to their levels in the pregnant cows, while NOMs (µmol/L) levels were significantly (*p* < 0.05) declined from day 38 in the same affected group with EED (Table 4), while no significant changes occurred in the control regarding both E2 and NOMs. There were group, time, and group*time interaction effects (*p* < 0.05, *p*-values were equal for all the effects).

## 4. Discussion

This study findings were considered an important indicator for luteal vascularization assessment in cows suffering from EED during heat stress with the help of hormonal analysis and luteal colored area calculation, as the reduction in luteal diameter, colored area, and progesterone levels at the 36–38 days after mating could be related to the death of the embryo (EED). To the best of our knowledge, this study is the first that assessed the relationship between luteal diameter, progesterone level, and luteal colored area that indicates amount of blood supply to the corpus luteum in relative to hemodynamic alterations in ipsilateral ovarian (OA) and uterine (MUA) arteries in both the pregnant normal group and another group with pregnancy loss due to EED in lactating cows. Cows in heat stress suffer from retardation in the intra-uterine environment with the marked reduction in the blood flow of the middle uterine artery at the ipsilateral side of the embryo in case of pregnancy; those alterations lead to increasing the chance of early embryonic death and suppression of the embryonic development due to elevation of the uterine temperature [33]. The effect of HS on reproductive performance is studied through several mechanisms, ether direct or indirect, as the direct one related to the reproduction and hormonal profile. 

In normal status, studies in pregnant cows and goats reported the changes in luteal vascularity [34,35]. These changes were expressed by elevation, especially in the ipsilateral side, and this could be due to the increased demand for nutrients in order to supply the newly developed embryo in association with progesterone levels from the corpus luteum [34]. Another study in heifers showed the marked increase in CL functionality and vascularity in the early pregnancy stage in relation to an elevation of the ipsilateral middle uterine artery [36]. Although this study was demonstrated on lactating cows, another study reported that all parts of CL (upper, middle, and lower) were higher in dry cows [22], while another study reported that the lactation status could affect the vascular perfusion [37]. The marked decline in both luteal diameter and luteal colored area from day 38 in another group with EED was in accordance with other studies reporting the same decline but from days 21 in non-pregnant beef cows [19] and non-pregnant beef heifers [38]. Previously, it was found that there is an increase in the luteal flow reported by Miyamoto et al. [39], followed by a decrease again due to prostaglandin emission of hours, which is an important vasoconstrictor. Furthermore, the elevation in circulating progesterone may affect the progesterone from CL during this phase, which increases the danger of pregnancy losses, especially in high-producing dairy cows under stress [40]. The marked growth and development of CL gravidities was associated with the angiogenesis process in the early CL as there was a loose organization in the capillaries network [41]. The negative relation that was recorded between both Doppler indices and Doppler peak velocity was similar to that recorded in other studies [42,43].

Our present study adds evidence for the connection between uterine and ovarian arteries, Doppler velocimetry, and embryonic growth by increasing the peak systolic velocity in normal cases or by decreasing the same velocity in EED; therefore, estimating uteroplacental blood circulation via Doppler is an important tool for evaluation of feto-maternal circulation [44]. Similarly, according to Scotti et al. [45], the uteroplacental artery PSV and EDV were elevated in queens during the gestation period. Conversely, other investigations reported no alteration in those blood flow parameters in the animal during pregnancy [46]. However, in the present study, there was a clear increase in ovarian and uterine PSV in the pregnant group, as indicated by high uterine vascular perfusion and high nutrient supply to the embryo. The blood flow within both uterine and ovarian arteries was associated with lowering the vessel resistance with elevation of the perfusion during the first months of gestation [46].

This study reported that both Doppler indices (RI and PI) were elevated in both MUA and OA from day 38 until day 50 after mating in cows with EED, which indicates a very low vascular perfusion reached to the uterus. Similar studies in cattle reported that due to an increase in demand of blood by the growing embryo, the RI and PI values decreased continuously during the gestation months and, thereafter, the values remained relatively constant until calving [46,47].This verified outcome supports prior research, which found a higher PI in uterine and ovarian arteries during the luteal phase in infertile women compared to fertile women [48,49]. Another study [50] found elevated vascular resistance in the uterine and ovarian arteries of infertile individuals, possibly caused by perivascular fibrosis originating from infection or endometritis, as well as the release of vasoactive mediators like prostaglandins.

This discovery holds greater significance in predicting the fate of future embryos derived from the same animals, since other research [51] discovered that the mean uterine arterial RI and PI values were lower in the pregnant group prior to embryo transfer than in the non-pregnant group. The significant decrease in vascular RI suggests that the vasculature distal to the evaluation location has less resistance to blood flow, which entails an increase in blood supply [52].

Estradiol causes a reduction in the systemic circulation vascular resistance due to its vasodilator effect when administered to women [53,54,55]. Moreover, it causes enlargement and dilatation of the uterine arteries in women [56]. Progesterone plays a role in opposing the effect of estradiol in some studies [56,57], but not in others [58]. The current study recommends that increasing estradiol and progesterone levels in the pregnant group might be related to increased uterine and ovarian luteal blood flow. Since implantation requires the uterine vasculature to respond maximally to progesterone and estradiol [59], their reduction in pregnancy loss is linked to a reduction in blood flow, which delays implantation and impairs the survival of the embryo. In fact, it is now believed that nitric oxide mediates the vasodilatation of uterine and systemic arteries induced by estradiol [60] and that nitric oxide production inhibitors counteract this effect [61]. Doppler ultrasound investigations seem to verify the current study’s findings that progesterone and estradiol are the cause of increased uterine blood flow in the early weeks of pregnancy [62]. The estradiol level decreased in the group that experienced pregnancy loss, which is consistent with prior research showing that lower E2 levels early in IVF pregnancies are linked to worse pregnancy outcomes for women. To forecast the likelihood of a live birth, estradiol can be utilized either by itself or in conjunction with hCG levels [63]. Nitric oxide (NO), produced by the endothelium of blood vessels, plays a role in the regulation of blood flow. It was identified in the placenta of sheep [64], and its level increased with advancing pregnancy under the effects of estradiol-17β, which is similar to our current study [65]. In addition, nitric oxide levels increased with gestational age in pigs [66]. Some studies reported that NO has a critical role in implantation and menstruation [67].

NO may be a critical vulnerability factor that regulates an individual’s risk of early pregnancy loss because, in line with our findings in the pregnancy loss group, women in the pregnant patient group had statistically significant decreased NO levels, which were linked to an increased risk for idiopathic recurrent miscarriage (*p* = 0.001), while elevated NO levels were measured in the normal pregnant women, non-pregnant patient group (*p* = 0.004).

## 5. Conclusions

This study provided novel facts about the relations among the luteal diameter, progesterone levels, and luteal hemodynamics to detect the alterations in the luteal function during the first 50 days after mating.

## Figures and Tables

**Figure 1 vetsci-11-00479-f001:**
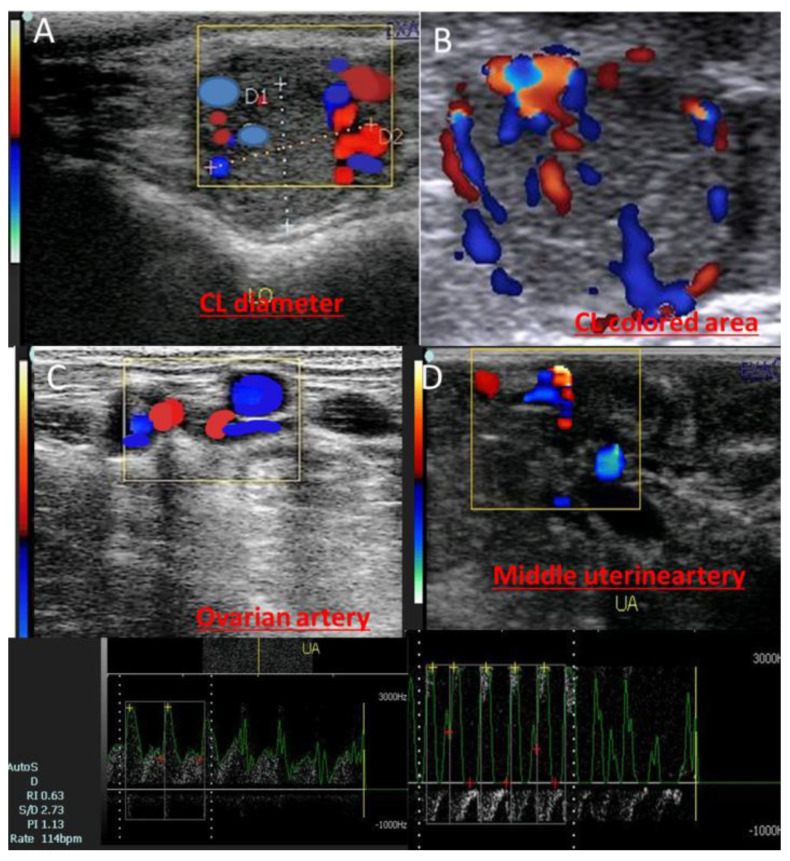
Ultrasongorams showed the corpus luteum of pregnancy (CL diameter; (**A**)), CL colored area measured by pixels via activation of the colored mode (**B**), in addition to the ovarian artery (**C**) and middle uterine artery (**D**) of the pregnant cows at day 20.

**Figure 2 vetsci-11-00479-f002:**
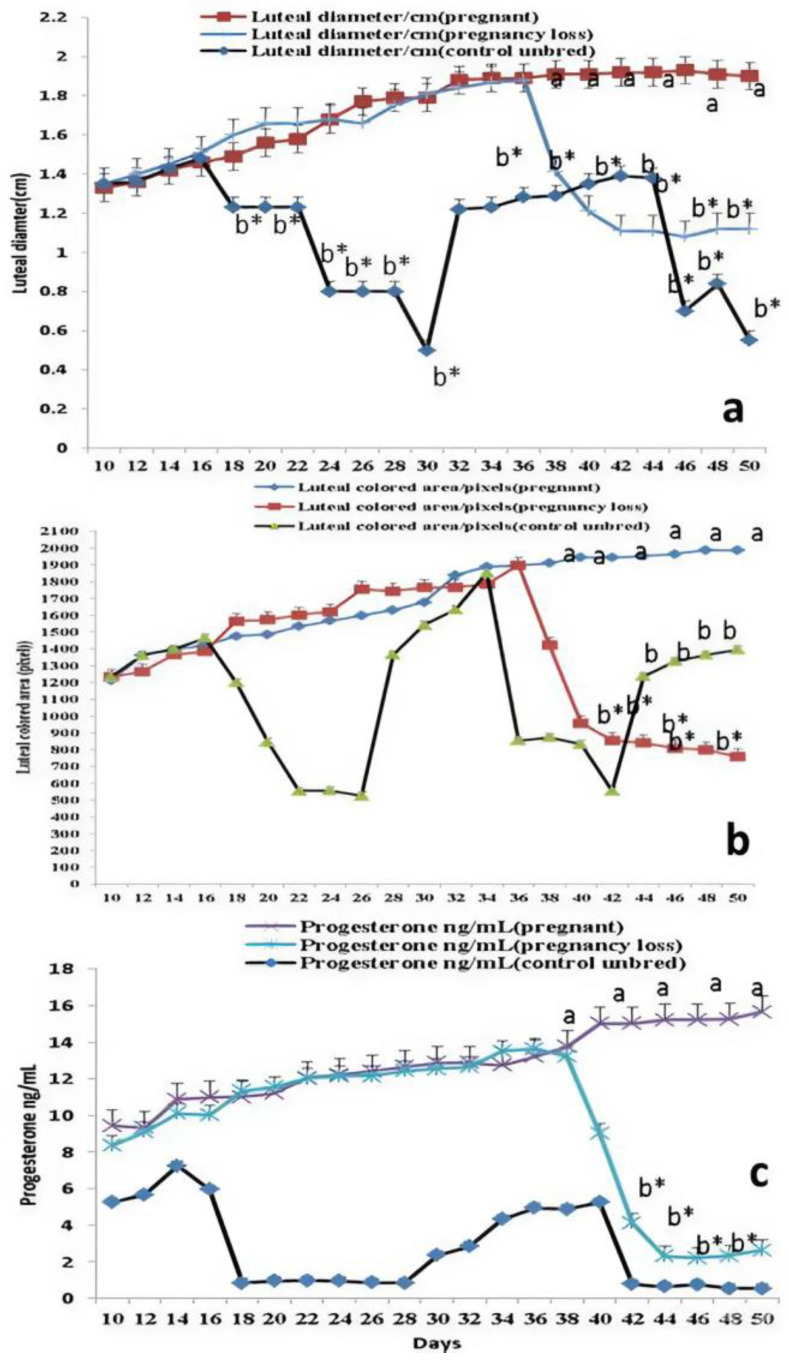
The luteal diameter (cm; (**a**)), luteal colored area (pixel; (**b**)), and levels of progesterone (ng/mL; (**c**)) from day 10 until day 50 after natural mating in control (not bred), pregnant, and pregnancy loss due to early embryonic death (EED) in cows. Data are represented as mean ± SEM. ^a^ and ^b^ values are significantly different at *p* < 0.05 compared with the control and pregnancy loss cows along the time points, while * value means that data are significantly different at *p* < 0.05 between the groups at the same time point.

**Figure 3 vetsci-11-00479-f003:**
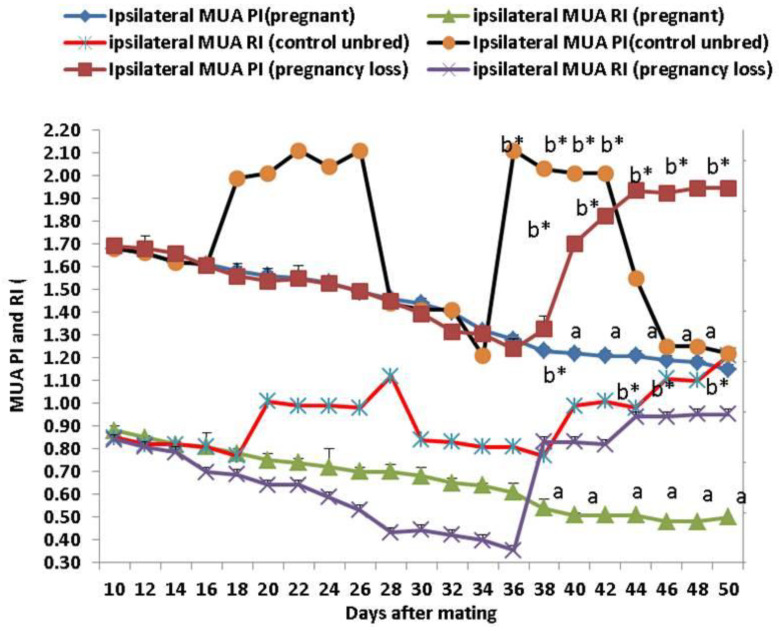
The middle uterine artery (MUA) Doppler indices expressed by resistance index (RI) and pulsatility index (PI) from day 10 until day 50 post natural mating in control (not bred), pregnant, and pregnancy loss due to early embryonic death (EED) in cows. Data are represented as mean ± SEM. ^a^ and ^b^ values are significantly different at *p* < 0.05 compared with the control and pregnancy loss cows along the time points, while * value means that data are significantly different at *p* < 0.05 between the groups at the same time point.

**Figure 4 vetsci-11-00479-f004:**
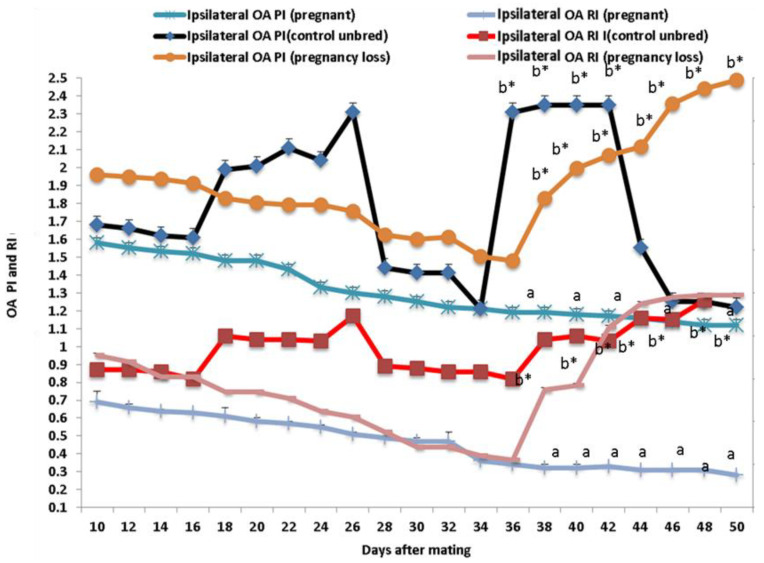
The ovarian artery (OA) Doppler indices expressed by resistance index (RI) and pulsatility index (PI) from day 10 until day 50 post natural mating in in control (not bred), pregnant, and pregnancy loss due to early embryonic death (EED) in cows. Data are represented as mean ± SEM. ^a^ and ^b^ values are significantly different at *p* < 0.05 compared with the control and pregnancy loss cows along the time points, while * value means that data are significantly different at *p* < 0.05 between the groups at the same time point.

**Figure 5 vetsci-11-00479-f005:**
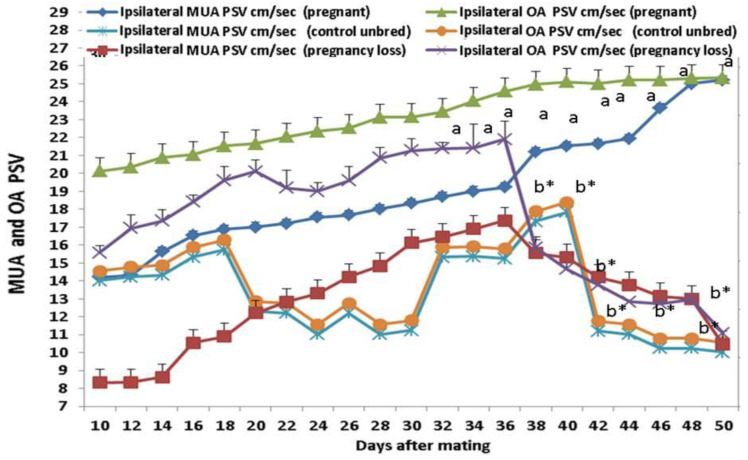
The ovarian artery (OA) and middle uterine artery (MUA) peak systolic velocity point of contraction (PSV; cm/sec) from day 10 until day 50 post natural mating in control (not bred), pregnant, and pregnancy loss due to early embryonic death (EED) in cows. Data are represented as mean ± SEM. ^a^ and ^b^ values are significantly different at *p* < 0.05 compared with the control and pregnancy loss cows along the time points, while * value means that data are significantly different at *p* < 0.05 between the groups at the same time point.

**Table 1 vetsci-11-00479-t001:** Calculations of temperature humidity index (THI) in relation to both temperature (T) and relative humidity (RH) along all months of the year 2022.

Months	T	RH	THI
Nov.	23.21	57.25	69.8
Dec.	22.31	54.52	67.5
Jan.	21.01	55.62	66.6
Feb.	21.14	56.32	66.9
Mar.	24.25	44.01	69.5
Apr.	25.32	44.5	71.2
May.	26.36	46.2	71.8
Jun.	38.11	51.25	88.9
Jul.	39.25	53.62	91.1
Aug.	42.21	61.25	97.11
Sep.	39.52	62.01	93.25
Oct.	23.21	56.25	68.25

**Table 2 vetsci-11-00479-t002:** Mean ± standard error of mean (SEM) of pregnancy-specific protein B (PSPB) levels in the pregnant and pregnancy loss groups from day 10 until day 50 after mating.

Days	PSPB (ng/mL)	
	P	EED
10	20.14 ^a^ ± 1.32	25.32 ^a^ ± 3.25
12	22.22 ^ab^ ± 7.52	28.24 ^ab^ ± 4.62
14	25.21 ^ab^ ± 8.25	28.52 ^ab^ ± 11.65
16	23.41 ^ab^ ± 3.45	29.32 ^ab^ ± 1.85
18	30.18 ^b^ ± 1.85	34.25 ^b^ ± 1.02
20	32.62 ^b^ ± 6.95	36.52 ^b^ ± 1.66
22	33.47 ^bc^ ± 2.54	38.21 ^bc^ ± 2.32
24	33.63 ^bc^ ± 2.32	38.32 ^bc^ ± 1.62
26	38.14 ^c^ ± 1.63	40.52 ^c^ ± 1.22
28	38.12 ^c^ ± 1.83	41.22 ^d^ ± 2.51
30	38.45 ^c^ ± 1.15	41.54 ^d^ ± 2.45
32	40.62 ^cd^ ± 2.22	46.85 ^e^ ± 1.32
34	41.12 ^d^ ± 2.11	47.22 ^e^ ± 1.74
36	41.74 ^d^ ± 2.95	46.21 ^e^ ± 2.25
38	39.93 ^d^ ± 2.32	40.85 ^c^ ± 1.02
40	43.74 ^de^ ± 3.55	33.82 ^b^* ± 6.75
42	42.55 ^e^ ± 2.66	22.47 ^a^* ± 1.54
44	43.45 ^e^ ± 1.25	20.6 ^a^* ± 2.32
46	44.75 ^e^ ± 5.26	18.25 ^a^* ± 4.02
48	46.12 ^f^ ± 1.76	19.22 ^a^* ± 1.45
50	47.12 ^f^ ± 1.25	19.74 ^a^* ± 1.55
*p*-value	0.03	0.03

Means with different superscripts within the column are significantly different at *p* < 0.05. * means there was a difference within the row at the same time points. P = pregnant; (EED) pregnancy loss due to early embryonic death.

**Table 3 vetsci-11-00479-t003:** Correlations between ipsilateral ovarian and middle uterine artery resistance index and pulsatility index in the pregnant and pregnancy loss due to early embryonic death at day 50 after mating.

Parameter	Ipsi MUA PI(EED)	Ipsi MUA RI(P)	Ipsi MUA RI(EED)	Ipsi OA PI(P)	Ipsi OA PI(EED)	Ipsi OA RI(P)	Ipsi OA RI(EED)
Ipsi MUA PI(P)	−0.862 **	0.391 **	−0.672 **	0.491 **	−0.55 **	0.66 **	−0.74 **
Ipsi MUA PI(EED)		−0.581 *	0.847 *	−0.473 *	0.39 *	−0.712 *	0.558 *
Ipsi MUA RI(P)			−0.369 **	0.547 **	−0.431 **	0.224 *	−0.411 *
Ipsi MUA RI(EED)				−0.324 *	0.732 *	−0.558 *	0.354 *
Ipsi OA PI(P)					−0.395 *	0.466 *	−0.265 *
Ipsi OA PI(EED)						−0.651 *	0.462 *
Ipsi OA RI(P)							−0.577 **

* The correlation is significant at the 0.05 level. ** The correlation is significant at the 0.01 level. (P) pregnant, (EED) pregnancy loss due to early embryonic death, Ipsi (ipsilateral), ovarian artery (OA), middle uterine artery (MUA), resistance index (RI), pulsatility index (PI), and peak systolic velocity (PSV).

**Table 4 vetsci-11-00479-t004:** Mean ± standard error of mean (SEM) estradiol and nitric oxide metabolite (NOM) levels in the control (not bred), pregnant, and pregnancy loss groups from day 10 until day 50 after mating.

Days		Estradiol (pg/mL)		NOMs (µmol/L)		
	C	P	EED	C	P	EED
10	81.54 ± 1.82	127.14 ^a^ ± 1.32	125.32 ^a^ ± 3.25	26.36 ± 3.25	45.23 ^a^ ± 2.31	46.32 ^a^ ± 2.63
12	83.22 ± 7.72	133.22 ^ab^ ± 7.52	138.24 ^ab^ ± 4.62	22.36 ± 6.25	49.32 ^ab^ ± 1.96	51.66 ^ab^ ± 2.84
14	88.21 ± 8.25	135.21 ^ab^ ± 8.25	134.52 ^ab^ ± 11.65	27.65 ± 4.25	51.32 ^ab^ ± 1.88	52.32 ^ab^ ± 2.93
16	88.41 ± 3.55	139.41 ^ab^ ± 3.45	142.32 ^ab^ ± 11.85	26.39 ± 1.25	52.32 ^ab^ ± 2.01	54.32 ^ab^ ± 1.88
18	78.48 ± 1.85	174.18 ^b^ ± 1.85	168.25 ^b^ ± 14.02	28.36 ± 2.14	52.66 ^ab^ ± 3.54	54.88 ^ab^ ± 1.36
20	97.82 ± 6.95	177.62 ^b^ ± 6.95	174.32 ^b^ ± 18.66	26.35 ± 2.05	52.48 ^ab^ ± 3.65	57.32 ^b^ ± 8.32
22	82.47 ± 1.54	182.47 ^bc^ ± 21.54	185.21 ^bc^ ± 21.32	24.65 ± 1.65	53.66 ^ab^ ± 1.65	56.77 ^b^ ± 7.32
24	88.36 ± 6.98	188.63 ^bc^ ± 28.32	192.32 ^bc^ ± 17.62	28.64 ± 1.22	58.31 ^bc^ ± 2.14	59.32 ^bc^ ± 4.62
26	75.62 ± 6.75	199.14 ^c^ ± 17.63	195.52 ^c^ ± 14.22	27.48 ± 1.25	63.12 ^c^ ± 1.05	62.32 ^cd^ ± 2.85
28	77.68 ± 7.11	201.12 ^c^ ± 11.63	207.25 ^c^ ± 18.62	28.65 ± 1.02	62.21 ^c^ ± 1.88	62.11 ^cd^ ± 2.36
30	85.41 ± 3.55	211.45 ^c^ ± 17.65	222.62 ^cd^ ± 17.21	27.10 ± 1.62	67.99 ^c^ ± 1.74	63.01 ^cd^ ± 4.12
32	88.48 ± 1.85	221.62 ^cd^ ± 18.22	228.85 ^d^ ± 10.32	23.65 ± 1.25	68.25 ^cd^ ± 1.21	64.33 ^cd^ ± 1.62
34	87.82 ± 6.95	235.12 ^d^ ± 20.31	250.22 ^ef^ ± 14.74	26.33 ± 1.66	68.32 ^cd^ ± 11.31	64.32 ^cd^ ± 6.85
36	82.67 ± 1.54	241.74 ^d^ ± 20.95	256.21 ^ef^ ± 16.25	25.32 ± 1.36	68.54 ^cd^ ± 3.65	51.66 ^d^* ± 7.32
38	98.86 ± 6.98	243.33 ^d^ ± 11.32	271.85 ^g^ ± 18.02	24.65 ± 1.25	71.62 ^d^ ± 4.22	47.57 ^a^* ± 6.66
40	88.55 ± 5.75	252.74 ^de^ ± 13.55	245.62 ^d^ ± 17.11	29.71 ± 5.22	71.88 ^d^ ± 6.82	47.12 ^a^* ± 4.62
42	77.36 ± 8.77	256.55 ^e^ ± 2.66	229.12 ^cd^ ± 18.22	28.33 ± 4.25	71.21 ^d^ ± 1.88	45.32 ^a^* ± 0.65
44	84.62 ± 8.78	258.45 ^e^ ± 11.25	210.21 ^c^ ± 14.25	26.95 ± 5.62	72.44 ^d^ ± 2.10	41.24 ^a^* ± 0.77
46	86.35 ± 9.78	259.75 ^e^ ± 15.26	133.25 ^ab^ ± 9.02	28.32 ± 2.14	72.36 ^d^ ± 4.01	41.95 ^a^* ± 1.21
48	74.32 ± 2.15	277.12 ^f^ ± 11.76	127.22 ^a^ ± 11.45	27.36 ± 1.32	78.32 ^e^ ± 0.54	30.14 ^a^* ± 0.88
50	77.98 ± 6.55	281.12 ^f^ ± 10.25	122.74 ^a^ ± 1.55	25.66 ± 5.26	79.55 ^e^ ± 1.99	30.02 ^a^* ± 0.74
*p*-value	0.87	0.01	0.01	0.68	0.01	0.01

Means with different superscripts within the column are significantly different at *p* < 0.05. * means there was a difference within the row at the same time points. *p* = pregnant.

## Data Availability

No new data were created due to privacy or ethical restrictions.

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
