# Peer review of "Vascular Alterations in Uterine and Ovarian Hemodynamics and Hormonal Analysis throughout Pregnancy Loss in Cows under Heat Stress"

_vetsci, 2024, doi:10.3390/vetsci11100479_

Round 1

Reviewer 1 Report

Comments and Suggestions for Authors

1.      Summary

The manuscript ‘Vascular Alterations in Uterine and Ovarian Hemodynamics, and Hormonal analysis throughout Pregnancy Loss in Cows under Heat Stress’ by Abdelnaby and collaborators aimed to examine the ovarian and middle uterine arteries blood flow and circulating concentrations of progesterone, estradiol, and nitric oxide metabolites in nonpregnant (nonbred) and pregnant cows and cows with early embryonic death under heat stress conditions. The authors have provided nice data on the hemodynamics of ovarian and uterine blood perfusion.

2.      Major concerns

The manuscript has major English problems. It is hard to understand the experimental design, results, and discussion which precludes thorough assessment of the study’s strengths. English grammar and the overall clarity of the text must be improved.

Major weakness include the description of experimental groups, data analysis, and clarity of the results (abstract, figures, figure legends, and text) that prevent my recommendation for publication of the manuscript in its current form.

Overall:

-        Clarification on the classification of cows in the EED group is imperative. How EED was determined to occur after day 38? Explain the criteria used to classify cows as EED. I suggest evaluating P4 and PSPB for each cow with pregnancy loss for better assessment of the day of pregnancy loss. EED likely occurred on different days, perhaps normalizing the data to the day relative to the event will bring useful information and strengthen the novelty of the manuscript. The following manuscripts may help with some insight on defining the day of pregnancy loss.  doi.org/10.3168/jdsc.2022-0282 and doi.org/10.3168/jds.2015-10192. Since you already analyzed PSPB, presenting some of these data (particularly comparing pregnant and EED) would strengthen the manuscript. You may also be able to compare the temporal relationship between changes in P4, PSPB, and vascular changes for the cows with pregnancy loss.

-        Data from nonbred cows after the end of their first estrous cycle (22-25-ish) should not be compared to pregnant and EED cows. It is irrelevant to compare the data from the second estrous cycle given the variability on estrous cycle length in lactating cows that would result on different days after estrus/ovulation. This is clearly observed with the progesterone profile for the first and second cycles. For the nonbred nonpregnant group, all data should only be presented until their ovulation at the end of the first estrous cycle.

-        Although cows were under heat stress due to environmental conditions at the time of the experiment (July-September), there were no comparisons to cows in a thermoneutral condition. Therefore, authors should be careful when suggesting the observed outcomes are due to heat stress. This statement is particularly relevant for the conclusions section. The authors should consider adding a statement clearly disclaiming that no comparisons were made between cows in heat stress and thermoneutral conditions.

-        The conclusions are an overstatement of the presented data not reflecting inferences drawn from the results found in the study. I suggest focusing the conclusions on the actual data evaluated in the study.

-        What about bred but nonpregnant cows? Do all bred cows become pregnant or had EED? This is unlikely as most pregnancy loss in lactating dairy cows occur before day 32. The authors should consider using PSPB (and P4) data to evaluate pregnancy loss before day 30 when embryonic heartbeat was assessed for pregnancy diagnosis.

Data analysis and figure presentation:

-        Description of statistical analysis is insufficient for data interpretation. Were the main effects of group (nonpregnant, pregnant, EED), day, and group by day interaction evaluated? P-values should be shown in figures or on legends. The description of symbols used in the figure must be improved. It is hard to understand what is being compared for a,b and *.

-        Figures with different y axis. The range of values for RI, PI, and PSV (left Y axis on figures 2, 3, and 4) for the pregnancy loss group is contained on the range of values for the other groups. Separating the Y axis for the groups makes it virtually impossible to assess the data. Please use the same Y axis for all groups for figures 2, 3, and 4.

-        Estradiol assay. The absolute values for estradiol for all groups are completely off compared to what has been reported for cows from other research groups (see doi.org/10.2527/jas.2012-5611, doi.org/10.1093/jas/skx043, doi.org/10.1093/biolre/ioaa065, doi.org/10.1530/REP-20-0362, doi.org/10.1016/S0093-691X(99)00182-X). The reference (29) for the estradiol and progesterone assay is from a manuscript from the same lab but there is no report of either estradiol or progesterone assay in the cited manuscript. Please provide details about the assay and justification whether the abnormal values for estradiol persist.

-        Why are estradiol and NOM data shown on a table? A figure could be useful for better visualization of the data, like in figure 1 for progesterone.

-        Correlation analysis. What data was used for the correlation analysis? The average of all evaluated days? Or a single day for the analysis? Please include how data were analyzed in the manuscript. Correlation analysis should only be performed within a group. Comparisons between groups do not make biological sense.

3.      Specific comment

-        Please avoid using words like ‘abnormal group’ and ‘normal’ as they appear to have different meanings throughout the text.

-        What does ‘CL gravidity’ mean?

-        ‘In pregnant cows, luteal diameter (cm) increased linearly from day 10 until day 48’ Please change the sentence as it is ambiguous. Does it mean it had a significant increase from day to day? Looking at the graph it does not seem likely.

-        Progesterone data (figure 1). It is not typical to have different concentrations of progesterone between pregnant and nonpregnant cows (before luteolysis in the nonpregnant). Do you have any insights for it or any justification? Same for estradiol (table 1). Were all nonbred cows cycling normally? Did they have any reproductive problems? Were all cows in the study in similar days in milk? Line 81 you say 12 multiparous lactating Holstein cows; what about the other six?

-        When you say ‘both groups’ which groups are you talking about?

-        You did not assess embryonic growth. Please revisit overstatement of data.

-        It is very hard to understand how pregnant and pregnancy loss groups were obtained. It appears that only 6 cows were pregnant on day 30 but later it appears to have 6 pregnant and 6 EED cows. Please clarify.

-        Early embryonic death is not an appropriate term to use as fetal period begins around 9 weeks after fertilization. Therefore, pregnancy loss after day 38 may be more adequately described as late embryonic death or simply as pregnancy loss.

-        References need to be provided for how CL blood flow was used for pregnancy diagnosis. Section 2.2

-        PSPB concentrations naturally decline in all pregnant cows between days 30 to 60-90. How was the decline in PSPB used to determined EED?

-        Were plasma or whole blood samples used for hormonal assays? Specific assay kits need to be provided for each assay.

Comments on the Quality of English Language

The manuscript has major English problems. It is hard to understand the experimental design, results, and discussion which precludes thorough assessment of the study’s strengths. English grammar and the overall clarity of the text must be improved.

Author Response

Reviewer 1

Summary

Comment 1: The manuscript ‘Vascular Alterations in Uterine and Ovarian Hemodynamics, and Hormonal analysis throughout Pregnancy Loss in Cows under Heat Stress’ by Abdelnaby and collaborators aimed to examine the ovarian and middle uterine arteries blood flow and circulating concentrations of progesterone, estradiol, and nitric oxide metabolites in nonpregnant (nonbred) and pregnant cows and cows with early embryonic death under heat stress conditions. The authors have provided nice data on the hemodynamics of ovarian and uterine blood perfusion.

 Response; thank you for your valuable opinion about our work we appreciated your comments and all the suggestions recommended by you are stated and well addressed

  1. Major concerns

Comment 2: The manuscript has major English problems. It is hard to understand the experimental design, results, and discussion which precludes thorough assessment of the study’s strengths. English grammar and the overall clarity of the text must be improved.

Major weakness include the description of experimental groups, data analysis, and clarity of the results (abstract, figures, figure legends, and text) that prevent my recommendation for publication of the manuscript in its current form.

 Response; thanks a lot for your recommendation about English problems we agree with your comments as many lines were revised to be more clear in English language as well as in typing with grammar in all the text :

Lines 47-55: this paragraph was revised for English problems and now was clear as suggested

Lines 72-82 , lines288-299,lines 311-317, as well as lines 333-343 were all revised for English language as suggested by the reviwer

Many parts of discussion as reorganized as well as the conclusion section is written again

Overall:

-        Clarification on the classification of cows in the EED group is imperative. How EED was determined to occur after day 38? Explain the criteria used to classify cows as EED. I suggest evaluating P4 and PSPB for each cow with pregnancy loss for better assessment of the day of pregnancy loss. EED likely occurred on different days, perhaps normalizing the data to the day relative to the event will bring useful information and strengthen the novelty of the manuscript. The following manuscripts may help with some insight on defining the day of pregnancy loss.  doi.org/10.3168/jdsc.2022-0282 and doi.org/10.3168/jds.2015-10192. Since you already analyzed PSPB, presenting some of these data (particularly comparing pregnant and EED) would strengthen the manuscript. You may also be able to compare the temporal relationship between changes in P4, PSPB, and vascular changes for the cows with pregnancy loss.

Response; thank you for your valuable comment , we agree with your comments but we would like to explain how we get cows with EED  as mentioned in lines 125-129 (From all mated cows, only six were became pregnant by evaluating the CL of pregnancy from day 10 till day 22 (n=6) with heartbeat at day 30, but some animals showed pregnancy loss and suffered from early embryonic death after day 38(EED, n = 6) that confirmed the decline in the plasma levels of pregnancy specific protein B (PSPB) that confirms the early death of the conceptus) those were confirmed also bu decline in progesterone levels as mentioned in (figure 1) in addition we add a table describe the levels of pregnancy specific protein B (PSPB) in both pregnant and non-pregnant cows to confirm our data as we already measured it in our study (Table 2)

-        Data from nonbred cows after the end of their first estrous cycle (22-25-ish) should not be compared to pregnant and EED cows. It is irrelevant to compare the data from the second estrous cycle given the variability on estrous cycle length in lactating cows that would result on different days after estrus/ovulation. This is clearly observed with the progesterone profile for the first and second cycles. For the nonbred nonpregnant group, all data should only be presented until their ovulation at the end of the first estrous cycle.

Response: thank you for your comments,we agree totally with this suggestion, but we need the unbred normal cows data to show how the blood flow parameters expressed by Doppler parameters of both ovarian and uterine arteries changed, as the study must contain a control group therefore we used unbred cows as a control group with no changes occurred in it .

-        Although cows were under heat stress due to environmental conditions at the time of the experiment (July-September), there were no comparisons to cows in a thermoneutral condition. Therefore, authors should be careful when suggesting the observed outcomes are due to heat stress. This statement is particularly relevant for the conclusions section. The authors should consider adding a statement clearly disclaiming that no comparisons were made between cows in heat stress and thermoneutral conditions.

Response; we agree as we now added a clear statement  (line 133-135)revealed that no comparison was made between cows in heat stress and thermo neutral conditions. But we add also the month by month changes in THI as suggested by other reviewer as we took the heat stress conditions as only a factor to show how hemodynamics were affected by this condition.

-        The conclusions are an overstatement of the presented data not reflecting inferences drawn from the results found in the study. I suggest focusing the conclusions on the actual data evaluated in the study.

Response: thank you for your comment we are totally agree with your suggestion as this conclusion part is now summarized to be more clear

-        What about bred but nonpregnant cows? Do all bred cows become pregnant or had EED? This is unlikely as most pregnancy loss in lactating dairy cows occur before day 32. The authors should consider using PSPB (and P4) data to evaluate pregnancy loss before day 30 when embryonic heartbeat was assessed for pregnancy diagnosis.

Response: we are totally agree with your comment as we already examined the PSPB and P4 ,in addition we add a table describe the levels of pregnancy specific protein B (PSPB) in both pregnant and non-pregnant cows to confirm our data as we already measured it in our study (Table 2)

we have Eighteen pluriparous cows were examined, with 12 only were subjected to the natural mating as other six animals were not bred remain as control group. Therefore we have now 12 cows were mated with Pregnancy diagnosis was confirmed at day 30 by embryonic heartbeat and CL gravidities (n=6; Pregnant), but some animals (n = 6) showed early embryonic death (EED) that was confirmed with progesterone and PSPB.

Data analysis and figure presentation:

-        Description of statistical analysis is insufficient for data interpretation. Were the main effects of group (nonpregnant, pregnant, EED), day, and group by day interaction evaluated? P-values should be shown in figures or on legends. The description of symbols used in the figure must be improved. It is hard to understand what is being compared for a,b and *.

Response: we use (ANOVA) of repeated measurements to study the effect of the days on ovarian and uterine arteries hemodynamics in pregnant cows and those with early embryonic death (EED) using SPSS software (2007). The description of symbols used in the figure is improved. As a, and b values are significantly different at P<0.05 compared with the control and pregnancy loss cows along the time points, while * value means data is significantly different at P<0.05 between the groups at the same time point. We cannot enter many alphabetical supercipts due to many days with large lines in the same figure and we could not separate each parameter in one figure because this will lead to presence of many figures in the paper.

-        Figures with different y axis. The range of values for RI, PI, and PSV (left Y axis on figures 2, 3, and 4) for the pregnancy loss group is contained on the range of values for the other groups. Separating the Y axis for the groups makes it virtually impossible to assess the data. Please use the same Y axis for all groups for figures 2, 3, and 4.

Response we have already used the same Y axis FOR FIGURE 2,3,AND 4 BUT ONE REALTED TO MUA and other related to OA ,we could not separate each parameter in one figure because this will lead to presence of many figures in the paper.

-        Estradiol assay. The absolute values for estradiol for all groups are completely off compared to what has been reported for cows from other research groups (see doi.org/10.2527/jas.2012-5611, doi.org/10.1093/jas/skx043, doi.org/10.1093/biolre/ioaa065, doi.org/10.1530/REP-20-0362, doi.org/10.1016/S0093-691X(99)00182-X). The reference (29) for the estradiol and progesterone assay is from a manuscript from the same lab but there is no report of either estradiol or progesterone assay in the cited manuscript. Please provide details about the assay and justification whether the abnormal values for estradiol persist.

Response : we add another refrences that stated the evalauation of both hormones and delete the previous old one (Ciernia LA, Perry GA, Smith MF, Rich JJ, Northrop EJ, Perkins SD, Green JA, Zezeski AL, Geary TW. Effect of estradiol preceding and progesterone subsequent to ovulation on proportion of postpartum beef cows pregnant. Anim Reprod Sci. 2021 Apr;227:106723. doi: 10.1016/j.anireprosci.2021.106723. Epub 2021 Feb 17. PMID: 33621845.)with all details about the assay Intra- and inter-assay coefficients of variation (CV) were 9 % and 20 %, respectively, for samples collected during the 3 years the study was conducted and 14 assays conducted to quantify E2. Assay sensitivity was 0.4 pg/ml. For measuring NOMs, serum samples were mixed with Griess reagent and incubated. The sensitivity of the assay was 0.225 µmol/L nitrites in the sample [31]. Serum PSPB levels was measured by radioimmunoassay method of double antibody. Assay sensitivity was 0.4 pg/ml for estradiol.

-        Why are estradiol and NOM data shown on a table? A figure could be useful for better visualization of the data, like in figure 1 for progesterone.

Response: thank you for your comment we need to reduce the amount of figure as there are too many figures in this paper therefore some data was presented in table to make many form of representation

-        Correlation analysis. What data was used for the correlation analysis? The average of all evaluated days? Or a single day for the analysis? Please include how data were analyzed in the manuscript. Correlation analysis should only be performed within a group. Comparisons between groups do not make biological sense.

 Response: we already make Correlation analysis within a group  but we collect two groups in same table to minimize the amount of tables present as Correlations between ipsilateral ovarian and middle uterine artery resistance index and pulstility index in the pregnant and pregnancy loss due to early embryonic death at day 50 after mating. we perfrom correlation to show positive and negative relations with each other specially Doppler parameters of arteries

  1. Specific comment

-        Please avoid using words like ‘abnormal group’ and ‘normal’ as they appear to have different meanings throughout the text.

Response: done as suggested

-        What does ‘CL gravidity’ mean?

Response: done as suggested CL gravidities = corpus luteum of pregnancy

-        ‘In pregnant cows, luteal diameter (cm) increased linearly from day 10 until day 48’ Please change the sentence as it is ambiguous. Does it mean it had a significant increase from day to day? Looking at the graph it does not seem likely.

Response: done as suggested

-        Progesterone data (figure 1). It is not typical to have different concentrations of progesterone between pregnant and nonpregnant cows (before luteolysis in the nonpregnant). Do you have any insights for it or any justification? Same for estradiol (table 1). Were all nonbred cows cycling normally? Did they have any reproductive problems? Were all cows in the study in similar days in milk? Line 81 you say 12 multiparous lactating Holstein cows; what about the other six?

Response: we have different levels of p4 pregnant and non pregnant as in pregnant the progesterone was declined due to luteolysis in addition the levels of progesterone in the control non bred cows was very low compared to its levels in pregnant cows due to presences of CL gravividatis as mentioned before ,the non bred cows were retured to the normal cycle as those are not suffered from any disorders. yes 12 animals was subjected to mating as the other six were not bred (Cows were examined (n=18) by ultrasound and then the experimental design was as follows: cows served as relevant control in this study that was open (not bred; n=6) cows, to assess timing of luteal regression and vascular conditions in this group relative to the other groups, then the other cows (n=12)

-        When you say ‘both groups’ which groups are you talking about?

Response: control unbred and pregnant one

--        It is very hard to understand how pregnant and pregnancy loss groups were obtained. It appears that only 6 cows were pregnant on day 30 but later it appears to have 6 pregnant and 6 EED cows. Please clarify.

 Response: Until day 30 we get all cows are pregnant but we observed that at day 38 most important parameters related to the embryo was declined therefore those six cows was suffered from early embryonic death as there were a loss of pregnancy

-        Early embryonic death is not an appropriate term to use as fetal period begins around 9 weeks after fertilization. Therefore, pregnancy loss after day 38 may be more adequately described as late embryonic death or simply as pregnancy loss.

Response: according to this accurate paper : early Embryonic Death. Embryonic mortality is the loss of the conceptus, which occurs in the first 42-45 days of pregnancy, the period from conception to completion of the differentiation stage. Parmar, S., Dhami, A.J., Hadiya, K.K., Parmar, C. 2016. Early Embryonic Death in Bovines: An Overview. Raksha Technical Review. 6. 6-12.

-        References need to be provided for how CL blood flow was used for pregnancy diagnosis. Section 2.2

Response: we add a refrence proof that cl of pregnant cows could used for pregnancy diagnosis Alcázar JL, Laparte C, López-Garcia G. Corpus luteum blood flow in abnormal early pregnancy. J Ultrasound Med. 1996 Sep;15(9):645-9. doi: 10.7863/jum.1996.15.9.645. PMID: 8866447.

-        PSPB concentrations naturally decline in all pregnant cows between days 30 to 60-90. How was the decline in PSPB used to determined EED?

Response: we add the data regarding PSPB in table 1 as suggested

-        Were plasma or whole blood samples used for hormonal assays? Specific assay kits need to be provided for each assay.

 Response: Blood was collected from the jugular vein , then after centrifuge the serum samples were collected in order to make hormonal assaying , yes we use kits Specific for each assay

Comments on the Quality of English Language

The manuscript has major English problems. It is hard to understand the experimental design, results, and discussion which precludes thorough assessment of the study’s strengths. English grammar and the overall clarity of the text must be improved.

Response: we revised the manuscript well and many lines reorganized and corrected for English problems as mentioned before please accept us as we are not native speaker for English language and we do our best to revise the whole paper. The discussion section is now improved as suggested

Reviewer 2 Report

Comments and Suggestions for Authors

The manuscript entitled “Vascular Alterations in Uterine and Ovarian Hemodynamics, and Hormonal Analysis throughout Pregnancy Loss in Cows Under Heat Stress” has in my opinion many flaws that make it unsuitable for publication. Specifically, the study has a very small sample size (n=6 per group) without providing a sound justification for that. Such small sample sizes make every finding questionable, as any variation with an animal or two will affect group values. Heat stress is referred to in the title, however, no comparison with non-stressed animals was performed. As a result, the lesser cannot draw any safe conclusion about the heat stress effect. The authors also do not report how they performed the numerous post-hoc analyses, which leaves us with questions regarding type II errors. Moreover, the applied techniques, mainly regarding doppler measurements, have not been described adequately. Another major concern I have, regards the early embryonic death group. From personal clinical experience and based on bibliographical data, I cannot imagine how 12 cows that were bred all conceived (this I have never experienced), that no cow had a miscarriage up to day 38 (it is well established that embryonic mortality is at its highest, almost 40 %, during or before implantation, i.e. day 19-22) and how suddenly 50 % of pregnant cows lost their fetus on d 38 (from d 28 onwards we refer to it as late embryonic death or fetal death and not EED).  These findings are very hard for me to explain biologically.

Comments on the Quality of English Language

Unfortunately, the manuscript is very hard to follow regarding English editing. 

Author Response

The manuscript entitled “Vascular Alterations in Uterine and Ovarian Hemodynamics, and Hormonal Analysis throughout Pregnancy Loss in Cows Under Heat Stress” has in my opinion many flaws that make it unsuitable for publication. Specifically, the study has a very small sample size (n=6 per group) without providing a sound justification for that. Such small sample sizes make every finding questionable, as any variation with an animal or two will affect group values.

Response: Thank you for your kind evaluation but the sample size in not small , because each animal were evaluated at different time points as we have 6 animals but observed for many days .Therefore; 6 animals * 21 days =126 observations , as the sample size is not small.

Heat stress is referred to in the title, however, no comparison with non-stressed animals was performed. As a result, the lesser cannot draw any safe conclusion about the heat stress effect. The authors also do not report how they performed the numerous post-hoc analyses, which leaves us with questions regarding type II errors. Moreover, the applied techniques, mainly regarding doppler measurements, have not been described adequately.

Response; thank you for your valuable comment we know that HS reduces estrus-related behaviors and continues affects fertility such as prolonged daytime and low conception rate. The indirect effect would be an adaptation of cow during HS by reduced feed intake. Postpartum reproductive diseases are major stressors that contribute to impaired and reduced fertility in dairy cattle. Retained placenta, metritis, and clinical endometritis are the three major diseases encompassing after calving.  Therefore we measured the blood flow as well as hormonal changes with the aid of ultrasound. The effect of HS on the reproductive performance is multidimensional through several mechanisms ether direct reproductive effects or indirect metabolic and nutritional effects. Off course we performed post hoc analysis and get the probability as we mentioned in statically analysis , we do not need to compare between heat stressed and non-heat stressed animals as we only measure Doppler parameters in this period

 Another major concern I have, regards the early embryonic death group. From personal clinical experience and based on bibliographical data, I cannot imagine how 12 cows that were bred all conceived (this I have never experienced), that no cow had a miscarriage up to day 38 (it is well established that embryonic mortality is at its highest, almost 40 %, during or before implantation, i.e. day 19-22) and how suddenly 50 % of pregnant cows lost their fetus on d 38 (from d 28 onwards we refer to it as late embryonic death or fetal death and not EED).  These findings are very hard for me to explain biologically.

Response: Embryonic mortality is the loss of the conceptus, which occurs in the first 42-45 days of pregnancy, the period from conception to completion of the differentiation stage.

it’s a large farm composed of cows and off course some cows showed early embryonic death and other complete the normal gestation period,but we focused in our study on only cows with normal pregnancy compared to pregnancy loss group to decrease error and we prepare our future paper on another group that was suffered from early miscarriage at days 22-40 .Therefore, this study estimated whether ipsilateral ovarian or uterine artery vascular dynamics would recognize cows at risk of pregnancy loss due to the early death of the newly developing embryo. The hypothesis was that cows that abort after d 38 would have decreased CL perfusion detected by the increase in Doppler indices (suggestive of decreased uterine blood flow), and the decline in peak systolic velocity of both ovarian and uterine arteries.

Reviewer 3 Report

Comments and Suggestions for Authors

Animals are exposed to heat stress when the body temperature is higher than the optimal range specified for the normal activity because the total heat load is greater than the capacity for heat dissipation.

First of all, I would like to refer to one inconsistency in the manuscript that casts a shadow on this extremely extensive and high-quality work. The authors state "LINE 90- The assessment of animals was in the hottest months from July to September, 2022 with a temperature humidity index (THI) more than 72." It is necessary to consistently present meteorological data for the examined geographical region for all seasons. It is also necessary to determine when the cows calved the previous time, whether the calving was during heat stress or in a thermoneutral period, what is the health status (infectious diseases, status on the farm), whether the heat stress was expressed throughout the day or in the hottest part days. It is necessary to insert a lot of details regarding the ambient conditions and their importance for the purpose of this experiment that was carried out.

In addition to the above, in order to be able to claim that our results are related to heat stress, it is necessary to carry out the same research in a thermoneutral period. Only then can we assume that heat stress is the cause of our findings.

This is the biggest complaint in the manuscript and needs to be addressed by the authors in order for the manuscript to be accepted.

The overall reproductive function of a herd of dairy cows is often estimated by calculating pregnancy rate, i.e., the product of estrus detection rate (how many cows in estrus are detected in estrus by farm personnel) and conception rate (a misnomer but a measure of how many cows that are inseminated are diagnosed as pregnant). A pregnancy rate of 100% would mean that every cow eligible to be pregnant in a 21-day period becomes pregnant in that time. By this measure, the reproductive function of the heat-stressed dairy cow can be very low indeed. Multiple reproductive processes are impaired, including oocyte competence, embryonic growth, gonadotropin secretion, ovarian follicular growth, steroidogenesis, development of the corpus luteum, and uterine endometrial responses. The effect of HS on the reproductive performance is multidimensional through several mechanisms ether direct reproductive effects or indirect metabolic and nutritional effects. The direct effect on ovarian activity concerning steroid hormone synthesis or steroidogenesis reduces concentration of estradiol and alters progesterone concentration. Consequently, HS reduces estrus-related behaviors and continuingly affects fertility such as prolonged day open and low conception rate. The indirect effect would be an adaptation of cow during HS by reduced feed intake. Postpartum reproductive diseases are major stressors that contribute to impaired and reduced fertility in dairy cattle. Retained placenta, metritis, and clinical endometritis are the three major diseases encompassing after calving. Several risk factors are associated with the development of reproductive diseases. Parity, dystocia, abortion, gender of calf, twinning, stillbirths, prolapsed uterus, body condition score, NEB, hypo-calcemia, and other stressors can predispose cows to a greater risk of developing reproductive disease.

The authors have well detected and understood all the reproductive problems that occur in heat stress. That is why they measured ultrasound, endocrine and biochemical parameters in parallel. We highly recommend this concept.

The material and method are well described. It is necessary to describe in more detail the biological, health and productive characteristics of the cows in the experiment. This trial lacks detailed data on farm management, quality and composition of food, calculative values ​​of meals and the like. All of the above can in itself affect the reproductive efficiency, so it is necessary to describe all these experimental circumstances in great detail.

It is necessary to describe reproductive techniques on the farm, natural mating, artificial insemination, applied artificial insemination technology and all other data related to males, that is, to technology.

The laboratory methods used should be described in detail. It is not enough to refer to similar works and give the CV% of the method. Please complete the laboratory analysis with all valid data from pre-analytical to analytical and post-analytical phase. The reason is that hormones are very variable and unstable, their value depends on many factors and it is necessary to look at all significant aspects and limitations during the expert reading of the manuscript.

The charts are clear.

Tables 2 and 3 are not technically adequately prepared and must be edited. The content is clear and ok.

The lack of typical ultrasound images with labeled surfaces and diameters is another shortcoming of this work. You need to present a large number of well-chosen images that clearly show morphological changes, pixel density, arrangement of cells and surfaces. This work is also methodologically interesting to a large extent, so such proofs and visualizations are necessary for future readers.

The discussion is well written. However, in the discussion itself, it is necessary to place the obtained results in the context of heat stress to a greater extent. All aspects of the physiology and pathology of reproduction are described in detail and quality, however, the context with heat stress is missing.

References are well chosen, but need to be technically arranged according to MDPI standards.

In general, this manuscript due to the concept and abundant research should be accepted with major changes and additions concerning: - supplementing with details related to meteorological conditions, THI index over a long period of time, farm management, laboratory analyzes and visualization of ultrasound findings, contextualization of the whole of work according to heat stress.

It would be ideal if the authors had the same experiment in the thermoneutral period. They would get valuable and indispensable results. If there are none, I encourage you to repeat the same experiment in the thermoneutral period.

Author Response

Reviewer 2:

  1. Animals are exposed to heat stress when the body temperature is higher than the optimal range specified for the normal activity because the total heat load is greater than the capacity for heat dissipation.

First of all, I would like to refer to one inconsistency in the manuscript that casts a shadow on this extremely extensive and high-quality work. The authors state "LINE 90- The assessment of animals was in the hottest months from July to September, 2022 with a temperature humidity index (THI) more than 72." It is necessary to consistently present meteorological data for the examined geographical region for all seasons.

Response: thank you for your evaluation of the paper ,first of all we are totally agree with your explanation and we are already added the data related to present meteorological data all the year 2022. We agree as we now added a clear statement (line 133-135) revealed that no comparison was made between cows in heat stress and thermo neutral conditions. But we add also the month by month changes in THI as suggested by other reviewer as we took the heat stress conditions as only a factor to show how hemodynamics were affected by this condition.

It is also necessary to determine when the cows calved the previous time, whether the calving was during heat stress or in a thermoneutral period, what is the health status (infectious diseases, status on the farm), whether the heat stress was expressed throughout the day or in the hottest part days. It is necessary to insert a lot of details regarding the ambient conditions and their importance for the purpose of this experiment that was carried out.

In addition to the above, in order to be able to claim that our results are related to heat stress, it is necessary to carry out the same research in a thermoneutral period. Only then can we assume that heat stress is the cause of our findings.

Response: The gestation period of cows was 9 month ±10 days , but we focused in our investigation on the blood flow ovarian and uterine arteries in addition to CL blood flow that related to early days of pregnancy in cows and not related to thermal conditions. we took only the heat stress concentration as a factor affecting. Therefore, we do not need to repeat the same examination in anther condition as this is not the point of interest the main aim of the paper is to determine the blood flow parameters in addition to hormonal profile in normal cows versus pregnancy loss one.

This is the biggest complaint in the manuscript and needs to be addressed by the authors in order for the manuscript to be accepted.

Response: the gestation period of cows was 9 month ±10 days , but we focused in our investigation on the blood flow ovarian and uterine arteries in addition to CL blood flow that related to early days of pregnancy in cows and not related to thermal condition we took only the heat stress c0ndtion as a factor affecting. Therefore, we do not need to repeat the same examination in anther condition as this is not the point of interest the main aim of the paper is to determine the blood flow parameters in addition to hormonal profile in normal cows versus pregnancy loss one.

Regarding the health status we mentioned in the materials that no infectious or reproductive disease were present and the farm condition was a building with a shadowing area for eating and drinking with an grazing area . we add this statement (The cows' health is preserved by clean housing, clean water, balanced diet, and taking the appropriate precautions against common diseases).

The overall reproductive function of a herd of dairy cows is often estimated by calculating pregnancy rate, i.e., the product of estrus detection rate (how many cows in estrus are detected in estrus by farm personnel) and conception rate (a misnomer but a measure of how many cows that are inseminated are diagnosed as pregnant). A pregnancy rate of 100% would mean that every cow eligible to be pregnant in a 21-day period becomes pregnant in that time. By this measure, the reproductive function of the heat-stressed dairy cow can be very low indeed. Multiple reproductive processes are impaired, including oocyte competence, embryonic growth, gonadotropin secretion, ovarian follicular growth, steroidogenesis, development of the corpus luteum, and uterine endometrial responses. The effect of HS on the reproductive performance is multidimensional through several mechanisms ether direct reproductive effects or indirect metabolic and nutritional effects. The direct effect on ovarian activity concerning steroid hormone synthesis or steroidogenesis reduces concentration of estradiol and alters progesterone concentration. Consequently, HS reduces estrus-related behaviors and continuingly affects fertility such as prolonged day open and low conception rate. The indirect effect would be an adaptation of cow during HS by reduced feed intake. Postpartum reproductive diseases are major stressors that contribute to impaired and reduced fertility in dairy cattle. Retained placenta, metritis, and clinical endometritis are the three major diseases encompassing after calving. Several risk factors are associated with the development of reproductive diseases. Parity, dystocia, abortion, gender of calf, twinning, stillbirths, prolapsed uterus, body condition score, NEB, hypo-calcemia, and other stressors can predispose cows to a greater risk of developing reproductive disease.

The authors have well detected and understood all the reproductive problems that occur in heat stress. That is why they measured ultrasound, endocrine and biochemical parameters in parallel. We highly recommend this concept.

Response; thank you for your valuable comment we know that HS reduces estrus-related behaviors and continuingly affects fertility such as prolonged day open and low conception rate. The indirect effect would be an adaptation of cow during HS by reduced feed intake. Postpartum reproductive diseases are major stressors that contribute to impaired and reduced fertility in dairy cattle. Retained placenta, metritis, and clinical endometritis are the three major diseases encompassing after calving.  Therefore we measured the blood flow as well as hormonal changes with the aid of ultrasound  

The material and method are well described. It is necessary to describe in more detail the biological, health and productive characteristics of the cows in the experiment. This trial lacks detailed data on farm management, quality and composition of food, calculative values ​​of meals and the like. All of the above can in itself affect the reproductive efficiency, so it is necessary to describe all these experimental circumstances in great detail.

Response;Thank you for your comment ,we addresede the following(Eighteen lactating pluriparous Holstein adult cows (n = 18) of 7-9 years old, with body condition score (BCS ;4 ± 0.5), and weighing (600 ± 50 kg) kept under maintenance conditions of feeding and management in the faculty research farm for large animals. Cow’s maintenance requirements of nutrition are consisted of concentrated rations, green fodder (40 kg), and hay with free access to water) . The reproductive assesment of cows were done using an ultrasound in addtion to signs after mating.

It is necessary to describe reproductive techniques on the farm, natural mating, artificial insemination, applied artificial insemination technology and all other data related to males, that is, to technology.

Response; we agree with this comment the type of mating was addressed (natural mating line 124),regarding the male we addressed that ( line 125-126 using the adult mature excellent bull with proven fertility 10-12 hrs after the onset of standing estrous)

The laboratory methods used should be described in detail. It is not enough to refer to similar works and give the CV% of the method. Please complete the laboratory analysis with all valid data from pre-analytical to analytical and post-analytical phase. The reason is that hormones are very variable and unstable, their value depends on many factors and it is necessary to look at all significant aspects and limitations during the expert reading of the manuscript.

Response; thank you for your perfect evaluation we added the following (all hormones are measured in a process of correct labeling, correct sampling, and correct amount of sample with a perfect transport (pre-analytical phase), then all samples were measured with a correct selection of the test ,regarding the biosafety measures(analytical), finally data were recorded in a correct report with an interpretation (post –analytical phase)

The charts are clear.

Tables 2 and 3 are not technically adequately prepared and must be edited. The content is clear and ok.

Response; we tried to make tables clear

The lack of typical ultrasound images with labeled surfaces and diameters is another shortcoming of this work. You need to present a large number of well-chosen images that clearly show morphological changes, pixel density, arrangement of cells and surfaces. This work is also methodologically interesting to a large extent, so such proofs and visualizations are necessary for future readers.

Response; thank you we agree with you we add ultrasound image related to ovarian and uterine artery with colored area pixels intensity of corpus luteum see the paper figure 1(

Figure 1; Ultrasongorams showed the corpus luteum of pregnancy (CL diameter; A), CL colored area measured by pixels via activation of the colored mode (B), in addition to ovarian artery (C) and middle uterine artery (D) of the pregnant cows at day 20.

The discussion is well written. However, in the discussion itself, it is necessary to place the obtained results in the context of heat stress to a greater extent. All aspects of the physiology and pathology of reproduction are described in detail and quality, however, the context with heat stress is missing.

Response we added this paragraph; Cows in heat stress are suffered from retardation in the intra uterine environment with the marked reduction in the blood flow of the middle uterine artery at the ipsilateral side of the embryo in case of pregnancy, those alterations lead to increasing the chance of early embryonic death and suppression of the embryonic development due to elevation of the uterine temperature [32]. The effect of HS on the reproductive performance is studied through several mechanisms ether direct  or indirect ways as the direct one related to the reproduction and hormonal profile

References are well chosen, but need to be technically arranged according to MDPI standards.

Response: Done

In general, this manuscript due to the concept and abundant research should be accepted with major changes and additions concerning: - supplementing with details related to meteorological conditions, THI index over a long period of time, farm management, laboratory analyzes and visualization of ultrasound findings, contextualization of the whole of work according to heat stress.

Response: we would like to thank you for your effort in evaluating our paper We provide supplementing with details related to meteorological conditions with THI in table 1.we add all information details about farm management condtion, laboratory analyses, as well as the presentation of an ultrasound picture as suggested.we also focused on the main idea with blood flow alterations in the whole of work according to heat stress.

It would be ideal if the authors had the same experiment in the thermoneutral period. They would get valuable and indispensable results. If there are none, I encourage you to repeat the same experiment in the thermoneutral period.

Response: thank you for your comment we are totally agree , but now the cows are not available to repeat the same experiment in the thermoneutral period. As all cows in this period were enter in synchronization program according to the farm condition , but we will do this in the future aspects and write this point in study limitations ,again we thank you very much for your help and hope that the paper now is suitable for publication ..

Reviewer 4 Report

Comments and Suggestions for Authors

The data here seems good and is interesting, but the submission needs a great deal of editing for English.  It will just take an individual who speaks English as a first language to go through it carefully.  It could be done in an afternoon, and it needs to be done because at present this is hurting the quality of the submission.

1) The first sentence of the “Simple Summary” does not make any sense.  Do you mean “Progesterone, estradiol, and nitric oxide metabolites were examined in blood from the ovarian and middle uterine arteries of cows under heat stress”.?

2) Overall, the “Simple Summary” needs to be rewritten for English.

3) Again, the first line in the “Abstract” does not make sense.  See (!).

4) Line 29 – “gravidities”?

5) Lines 38-41 – There are lots of spacing errors in the text.  It is difficult for the reader to egnore such errors.

6) Line 59 – There is just a fragment of a sentence here.

7) Line 126 – Remove “excellent”.  It is enough to say “of proven fertility”.

8) Line 168 – Should probably state here what days of the cycle or pregnancy blood were collected.

9) There is not nearly enough information or detail in section 2.5.  This section must be properly written to address all of the blood assays and how they were performed.

10) Figure 2 – Why do you change the colors and symbols from one panel to the next?  All the pregnant should be one color and symbol, and the pregnancy loss should be another color and symbol, and all of the control unbred should be a third color and symbol.

11) Figures 3, 4 and 5 – You need to use either “ipsilateral” or Ipsilateral” in the figure.

12) Lines 363-368 - some references have a direct discussion of the ways nitric oxide influences implantation and early placentation in ruminants.  You might use some citations in that to fill out your Discussion.

13) This reviewer thinks the “Introduction” would benefit from a paragraph covering pregnancy loss in cattle in general. 

Comments on the Quality of English Language

The data here seems good and is interesting, but the submission needs a great deal of editing for English.  It will just take an individual who speaks English as a first language to go through it carefully.  It could be done in an afternoon, and it needs to be done because at present this is hurting the quality of the submission.

Author Response

The data here seems good and is interesting, but the submission needs a great deal of editing for English.  It will just take an individual who speaks English as a first language to go through it carefully.  It could be done in an afternoon, and it needs to be done because at present this is hurting the quality of the submission.

Response; Thank you for your comment. , we already get help for English by a native speaker to get the paper more clear. Thank you very much, The revised paper is now clear as suggested

  • The first sentence of the “Simple Summary” does not make any sense.  Do you mean “Progesterone, estradiol, and nitric oxide metabolites were examined in blood from the ovarian and middle uterine arteries of cows under heat stress”.?

Response; we mean that blood samples were collected and hormones were analyzed .We modified the sentence to be (The ovarian and middle uterine arteries blood flow were examined in cows under heat stress conditions regarding hormonal profile)

  • Overall, the “Simple Summary” needs to be rewritten for English.

Response; this part was totally written again as suggested: (The ovarian and middle uterine arteries blood flows were examined in cows under heat stress conditions regarding hormonal profile. Luteal vascularity was declined in cows with embryonic death. Progesterone levels elevated in cows with embryonic death then declined .In addition, Both Doppler indices were elevated in cows suffered from embryonic death. This study provided facts about the relations among the luteal diameter and luteal hemodynamics that predict the amount of blood supply which act as a sensitive parameter to detect the alterations in luteal function during the first 50 days after mating)

  • Again, the first line in the “Abstract” does not make sense.  See (!).

Response; modified as suggested

  • Line 29 – “gravidities”?

Response; corpus luteum of pregnancy is called CL graviditatis, spelling is corrected

  • Lines 38-41 – There are lots of spacing errors in the text.  It is difficult for the reader to egnore such errors.

Response. Many spacing errors are corrected as suggested

  • Line 59 – There is just a fragment of a sentence here.

Response: the sentence is now rewritten.

  • Line 126 – Remove “excellent”.  It is enough to say “of proven fertility”.

Response; removed.

  • Line 168 – Should probably state here what days of the cycle or pregnancy blood were collected.

Response; Blood was collected from the jugular vein from day 10to day 50 at early morning by 18 gauge needle

  • There is not nearly enough information or detail in section 2.5.  This section must be properly written to address all of the blood assays and how they were performed.

Response: we add catalogue number for assays, we add inter and intra assays coefficients, and we add all information regarding PSBP and nitric oxide.

  • Figure 2 – Why do you change the colors and symbols from one panel to the next?  All the pregnant should be one color and symbol, and the pregnancy loss should be another color and symbol, and all of the control unbred should be a third color and symbol.

Response;  we change colors because we have many parameters in one figure, as shown in figure 3 ,4, and 5 therefore, colored lines expressed by item shown in each figure are more clear than tables.

  • Figures 3, 4 and 5 – You need to use either “ipsilateral” or Ipsilateral” in the figure.

Response: done

  • Lines 363-368 - some references have a direct discussion of the ways nitric oxide influences implantation and early placentation in ruminants.  You might use some citations in that to fill out your Discussion.

Response We add a paragraph as suggested. Nitric oxide (NO), produced by the endothelium of blood vessels, plays a role in the regulation of blood flow was identified in the placenta of sheep[64] and its level increased with advancing pregnancy under the effects of estradiol-17β which is similar to our current study [65], in addition to nitric oxide levels increased with gestational age in pigs[66].some studies reported that NO has a critical role in implantation and menstruation [67]NO may be a critical vulnerability factor that regulates an individual's risk of early pregnancy loss because, in line with our findings in the pregnancy loss group, women in the pregnant patient group had statistically significant decreased NO levels, which were linked to an increased risk for idiopathic recurrent miscarriage (p=0.001), while elevated NO levels were measured in the normal pregnant women, non-pregnant patient group (p=0.004) [68]).

13) This reviewer thinks the “Introduction” would benefit from a paragraph covering pregnancy loss in cattle in general.

Response: there is already paragraph stated the pregnancy loss (Methods for increasing the number of calves and heifers produced have been developed [3], but fetal losses and the premature death of developing embryos provide an overwhelming obstacle. Cows at risk of miscarriage can be identified using a variety of techniques, which may present opportunities to develop intervention programs that will benefit those harmed [4]. The constant exposure to high humidity and warmth in the environment had an impact on steroidogenesis as well as oocyte viability [5].

Round 2

Reviewer 1 Report

Comments and Suggestions for Authors

The authors addressed all comments from my first review. However, the manuscript still presents several weaknesses on data analysis, description of the results, and overstatement of findings that prevents my recommendation for acceptance.

Please change the word “facts” on line 39.

Thank you for clarifying in the text that no comparisons were made between cows in heat stress and thermoneutral conditions. After this and other clarifications, it appears that cows became pregnant throughout the year and no descriptions were made about which cows were actually under heat stress when they became pregnant or had pregnancy loss. What does it mean that THI was taken as a factor?

Statistical analysis and results are inadequately described to understand findings. For example, in figures 2-5, what are the P values for an effect of group, time, and group*time interaction? In figure 1a, it is indicated (a-b superscript) that luteal diameter is different between control and pregnant cows on day ~34 to 50; what about days 24 to 30?

Hormonal analysis: there is still no reference for PSPB assay. It looks like you added a reference for estradiol and progesterone in cows instead of mares.  

There is no justification for the excessively high concentrations of estradiol.

 Information (Brand and catalog number) for all hormonal assays need to be provided.

The conclusion is still an overstatement of the data as there were no analysis comparing the effect of THI on pregnancy loss.

Figures with different y axis. Separating the Y axis for the groups (pregnant and control vs pregnancy loss) makes it virtually impossible to assess the data. How are readers compare any of the end-points between pregnant and pregnancy loss groups when different scales are used? Please use the same Y axis for all groups for figures 3-5.

Correlation analysis. What data was used for the correlation analysis? The average of all evaluated days? Or a single day for the analysis? Please include how data were analyzed in the manuscript. 

Comments on the Quality of English Language

Several sentences were edited/added but overall English grammar and the overall clarity of the text must be improved.

Author Response

The authors addressed all comments from my first review. However, the manuscript still presents several weaknesses on data analysis, description of the results, and overstatement of findings that prevents my recommendation for acceptance

 Response: Thank you for your evaluation of the paper. ,first of all we are totally agree with your explanation, and we have already added the data regarding data analysis as well as results descriptions. Please accept our revised version of the paper. we do our best to improve it .

Please change the word “facts” on line 39.

 Response: done, the word changed into data.

Thank you for clarifying in the text that no comparisons were made between cows in heat stress and thermoneutral conditions. After this and other clarifications, it appears that cows became pregnant throughout the year and no descriptions were made about which cows were actually under heat stress when they became pregnant or had pregnancy loss. What does it mean that THI was taken as a factor?

Response: we mentioned in materials that the examination was done in the summer so we take THI as factor for this period because the heat stress conditions affects negatively several reproductive processes, resulting in a pronounced depression of conception rate in dairy cows worldwide. Therefore; we took into consideration the monthly metrological data of this year as suggested by other reviewer.

Statistical analysis and results are inadequately described to understand findings. For example, in figures 2-5, what are the P values for an effect of group, time, and group*time interaction? In figure 1a, it is indicated (a-b superscript) that luteal diameter is different between control and pregnant cows on day ~34 to 50; what about days 24 to 30?

Response:  thank you for your comment We already added now the group effect, time effect, as well as their interaction in the results section regarding figures in lines 202-225 and 250. We   use repeated measures ANOVA by general linear model as the a, and b values are significantly different at P<0.05 compared with the control and pregnancy loss cows along the time points, which is group and time effect, while * value means data is significantly different at P<0.05 between the groups at the same time point. For figure 1a we added the superscript as suggested now but for other figures  days 24-30 off course changes are occurred but not significantly as days 34-50 therefore we focused on the significance effect.

Hormonal analysis: there is still no reference for PSPB assay. It looks like you added a reference for estradiol and progesterone in cows instead of mares.  

Response: we add references number 29 and 32

There is no justification for the excessively high concentrations of estradiol.

Response: this paragraph is present in the discussion ( Estradiol causes a reduction in the systemic circulation vascular resistance due to its vasodilator effect when administered to women [53,54, 55], moreover, it causes enlargement and dilatation of the uterine arteries in women[56], progesterone plays a role in opposing the effect of estradiol in some studies [56,57], but not in others[58]. The current study recommends that increasing estradiol and progesterone levels in the pregnant group might be related to increased uterine and ovarian luteal blood flow. Since implantation requires the uterine vasculature to respond maximally to progesterone and estradiol [59],

 Information (Brand and catalog number) for all hormonal assays need to be provided.

Response: K030-H is a catalog number for estradiol, 6107620 is for progesterone, and for PSPB the catalog number is CSB-E13353B (Competitive EIA)

The conclusion is still an overstatement of the data as there were no analysis comparing the effect of THI on pregnancy loss.

Response: this part was deleted as suggested; now this section is summarized as suggested

Figures with different y axis. Separating the Y axis for the groups (pregnant and control vs pregnancy loss) makes it virtually impossible to assess the data. How are readers compare any of the end-points between pregnant and pregnancy loss groups when different scales are used? Please use the same Y axis for all groups for figures 3-5.

Response: we make one Y-axis for figure 3 ,4, and 5  as suggested by the reviewer to make reading of the data more clear

Correlation analysis. What data was used for the correlation analysis? The average of all evaluated days? Or a single day for the analysis? Please include how data were analyzed in the manuscript. 

Response: we use single day of analysis and write this in the correlation. Correlations between ipsilateral ovarian and middle uterine artery resistance index and pulstility index in the pregnant and pregnancy loss due to early embryonic death at day 50 after mating.

Reviewer 2 Report

Comments and Suggestions for Authors

The authors have not improved their manuscript in any manner. They simply argued to my comments, without convincing me for their study whatsoever. The way they evaluate sample size is unique (sample size*times measured). No sample size software calculates sample size this way.

Comments on the Quality of English Language

Manuscript is in many cases incomprehensible. An editing service should be used in case the journal decides to accept your manuscript.

Reviewer 3 Report

Comments and Suggestions for Authors

Thank you for improvements. Please prepare reference list according to MDPI style.

Author Response

Thank you for improvements. Please prepare reference list according to MDPI style.

Response; thank you very much We corrected the references according to the journal style.

Reviewer 4 Report

Comments and Suggestions for Authors

The authors response is ok.